# The Effects of Microbial Additive Supplementation on Growth Performance, Blood Metabolites, Fecal Microflora, and Carcass Characteristics of Growing–Finishing Pigs

**DOI:** 10.3390/ani14091268

**Published:** 2024-04-23

**Authors:** Hyuk-Jun Lee, Bu-Gil Choi, Young-Ho Joo, Chang-Hyun Baeg, Ji-Yoon Kim, Dong-Hyeon Kim, Seong-Shin Lee, Sam-Churl Kim

**Affiliations:** 1Division of Applied Life Science (BK21Four, Institute of Agriculture and Life Science), Gyeongsang National University, Jinju 52828, Republic of Korea; hyukjun0209@gmail.com (H.-J.L.); vet00161@gmail.com (B.-G.C.); wn5886@gmail.com (Y.-H.J.); back5254@naver.com (C.-H.B.); in42869@naver.com (J.-Y.K.); 2Dairy Science Division, National Institute of Animal Science, Rural Development Administration, Cheonan 31000, Republic of Korea; kimdh3465@korea.kr; 3Animal Nutrition and Physiology Division, National Institute of Animal Science, Rural Development Administration, Wanju 55356, Republic of Korea; shin7398@korea.kr

**Keywords:** blood metabolite, carcass characteristic, growth performance, fecal microflora, microbial additive, pig

## Abstract

**Simple Summary:**

Dietary probiotic strategies that benefit animal performance by producing antibacterial substances in the intestine, competing with harmful gut flora, and stimulating the immune system should be developed. Thus, this study examined optimum levels of growing–finishing pigs using a mixture of microbial additives producing antimicrobial substances and digestive enzymes to improve growth performance, blood metabolites, fecal microflora, and carcass characteristics. Three treatments were used: 0, 0.5, and 1.0% microbial additives in the basal diet, which led to a higher average daily gain and feed efficiency in growing–finishing pigs but not in the initial and final weights. Supplementation of pig diets with microbial additives has been demonstrated to be an effective strategy for improving the conent of immunoglobulin G (IgG) as a blood metabolite, increasing fecal lactic acid bacteria count, and reducing *Escherichia coli* (*E. coli*) count in pig manure. However, the use of microbial additives to improve carcass characteristics has been questioned due to their lack of influence on pigs. Consequently, 1.0% microbial additive could be optimal for growing–finishing pigs to improve growth performance, IgG content, and the fecal microflora environment.

**Abstract:**

This study aimed to assess the effects of microbial additives that produce antimicrobial and digestive enzymes on the growth performance, blood metabolites, fecal microflora, and carcass characteristics of growing–finishing pigs. A total of 180 growing–finishing pigs (Landrace × Yorkshire × Duroc; mixed sex; 14 weeks of age; 58.0 ± 1.00 kg) were then assigned to one of three groups with three repetitions (20 pigs) per treatment for 60 days of adaptation and 7 days of collection. Dietary treatments included 0, 0.5, and 1.0% microbial additives in the basal diet. For growth performance, no significant differences in the initial and final weights were observed among the dietary microbial additive treatments, except for the average daily feed intake, average daily gain, and feed efficiency. In terms of blood metabolites and fecal microflora, immunoglobulin G (IgG), blood urea nitrogen, blood glucose, and fecal lactic acid bacteria count increased linearly, and fecal *E. coli* counts decreased linearly with increasing levels of microbial additives but not growth hormones and *Salmonella*. Carcass quality grade was improved by the microbial additive. In addition, carcass characteristics were not influenced by dietary microbial additives. In conclusion, dietary supplementation with 1.0% microbial additive improved average daily gain, feed efficiency, IgG content, and fecal microflora in growing–finishing pigs.

## 1. Introduction

Over the past 50 years, antibiotic-based growth promoters have been used in farm animal production in several countries. Dietary supplementation of farm animals with antibiotics improves growth productivity, disease prevention, and farm income [1,2]. However, the excessive use of antibiotics has notably impacted livestock and public health owing to problems such as residual livestock products, antibiotic resistance when ingesting residual livestock products, and the multiplication of pathogenic harmful bacteria. Therefore, growth-promoting antibiotics have been prohibited in animal feed in Europe since 2006 and in South Korea since 2011 [3,4,5]. Consequently, as there is increasing awareness of the potential negative effects of animal diets, there has been increased interest in producing livestock without using antibiotic growth promoters [6]. Many livestock producers have suggested the use of various antibiotics to enhance animal performance, disease prevention, health, and meat quality. One of the most effective strategies that has been successfully used to control these problems is microbial additives. Microbial additives are preparations or products containing defined concentrations of live microorganisms that are sufficient to alter the intestinal microflora of the host and exert beneficial health effects [7]. Microbial additive supplementation has been suggested to improve growth performance [8,9,10], the immune system [11,12], and fecal microflora [13,14]. Multi-strain or multi-species microbial additives have been found to be more effective than mono-strain or single-species additives [15]. For example, considering non-antibiotic feed additives in pig diets, the additives that are available for improving growth performance or the gut and fecal environment through inclusion in diets and include the believed mechanisms for each additive are classified into six primary categories: (i) acidifier, (ii) mineral, (iii) prebiotics, (iv) probiotics (direct-fed microbials, DFM), (v) nucleotides, and (vi) plant extracts, as described by Liu et al. [6]. More recently, probiotics and prebiotics have been used successfully in pig diets for several years. Firstly, probiotics are commonly known as direct-fed microbials and are considered “live microorganisms that confer a health benefit on the host when administered in adequate amounts [16]”. Prebiotics are non-digestible, fermented food substrates that stimulate growth, change the composition and activity of gut microorganisms, and improve host health [17]. These positive effects of probiotic and prebiotic supplementation may be worthwhile as feed additives for animals to be used in a way that has more benefits for animal health and performance, especially in growing to fattening phase situations and animals exposed to greater pathogenic loads [18]. Currently, the aim of the pig industry has focused on the accumulation of scientific evidence with respect to microbial additives, such as probiotics and prebiotics, and their effect on the growth, production, and health of pigs, as well as their effect on the immune system, digestive tract, and blood metabolites. In response to San Andres et al. [18], these changes were aimed at improving the ability of pigs to prevent pathogenic bacteria from colonizing the intestinal system, which can be accomplished via mechanisms that reduce the damaging effects of pathogens on the host. Microbial additives containing *Bacillus* spores improved the weight gain, feed conversion ratio, and carcass quality of the pigs [8]. In contrast, probiotics with the *Lactobacillus* strain can increase the gut immunoglobulin A (IgA) immune response and promote the gut immunological barrier [12], while *Saccharomyces* supplementation increased fecal lactic acid bacteria counts and pathogenic bacteria such as *Escherichia coli* (*E. coli*) decreased in number in pigs [13]. However, few attempts have been made to develop multi-microbe microbial additive products, and reports on the effects of multi-species microbial additives on growing–finishing pigs are limited.

Therefore, we hypothesized a positive influence of multi-microbe microbial additive products on the production and immune response of the blood metabolites, microflora, and meat quality of pigs. The objective of this study was to investigate the effect of microbial additives producing antimicrobial substances and digestive enzymes on the growth performance, blood metabolites, fecal microflora, and carcass characteristics of growing to finishing pigs.

## 2. Materials and Methods

The animal experiments were conducted at the Goseong Pig Farm (Gyeongnam, Republic of Korea) and were approved by the Animal Care and Use Committee of Gyeongsang National University, Jinju, Republic of Korea (GNU-200608-P0034).

### 2.1. Probiotics

The microbes included *Lactobacillus plantarum* SK3121 (>9.0 log10 colony-forming units (CFU)/g), *Bacillus subtilis* SK877 (>9.0 log10 CFU/g), *B. amyloliquefaciens* BBG-B5 (>9.0 log10 CFU/g), and *Saccharomyces cerevisiae* SK3587 (>9.0 log10 CFU/g), which were used as the seedstocks. *L. plantarum* SK3121 and *B. subtilis* SK877 were isolated based on their antimicrobial activity in Kimchi (Korean traditional fermented cabbage) and digestive enzyme activity in corn silage, respectively [4]. *B. amyloliquefaciens* BBG-B5 and *S. cerevisiae* SK3589 were isolated from pig feces based on their digestive enzyme activity, nutrients, and growth factors. The microbial additive used in this study, in which the seedstocks were applied into the grain mixtures at a 2% (as-fed basis) and ensiled at 30 °C for 7 days, was purchased from Big Biogen (Anseong, Republic of Korea). The counts of the microbial additive are presented in Table 1.

### 2.2. Animal Management

A total of 180 growing–finishing pigs (Landrace × Yorkshire × Duroc; mixed sex; 14 weeks of age; 58.0 ± 1.00 kg) were randomly divided into three treatments with three repetitions (20 pigs) per treatment for 60 days of adaptation and 7 days of collection. The dietary treatments consisted of 0% (basal diet), 0.5% (basal diet + 0.5% microbial additive), and 1.0% (basal diet + 1.0% microbial additive). The basal diet was used throughout the experimental period (Table 2). The pens were fully slatted with concrete panels, and the light and temperature conditions were automatically controlled. Pigs were fed ad libitum using a one-hole feeder in each pen. The diet was delivered twice daily at 09:00 h and 17:00 h, and water was provided ad libitum per pen via nipples.

### 2.3. Analysis

#### 2.3.1. Diet Chemical Composition

The feed (1 kg) was dried at 65 °C for 48 h in a forced-air oven and ground using a cutting mill to pass through a 1 mm screen (Shinmyung Electric Co., Ltd., Gimpo, Republic of Korea). The metabolizable energy in the feed was calculated using the energy values of the ingredients obtained from the NRC [19]. The dry matter concentration was determined using a forced-air drying oven at 105 °C for 24 h. Crude protein and ether extracts were measured according to the Kjeldahl method (method number 984.13; AOAC, [20]) and the Soxhlet method (method number 920.39; AOAC, [20]), respectively. Crude ash was determined via incineration at 550 °C for 4 h in a muffle furnace.

#### 2.3.2. Microbial Counts

The microbial additive sample (20 g) was placed in 180 mL of distilled water and processed in a blender for 30 s. The extract was filtered through two layers of cheesecloth and diluted (10^−6^ to 10^−8^) to determine the microbial counts for lactic acid bacteria (LAB), bacilli, and yeast [21,22]. Microbial counts were measured via plate counting on Lactobacilli Man Rogosa Sharpe agar (MRS; Difco, Detroit, MI, USA) for LAB, Luria-Bertani agar (LB; Difco, Detroit, MI, USA) for *Bacillus* and potato dextrose agar (PDA; Difco, Detroit, MI, USA) for yeast. The MRS agar plates were maintained in a CO_2_ incubator (Thermo Scientific, Waltham, MA, USA) at 30 °C for 48 h. The LB agar and PDA plates were incubated for 48 h at 30 °C under an aerobic incubator (Johnsam Corp., Boocheon, Republic of Korea) [23]. Visible colonies on the plates were counted and expressed as colony-forming units (log10 CFU/g of the sample).

#### 2.3.3. Growth Performance

To analyze growth performance, each pig was weighed at the beginning (day 1) and end (day 60) of the experimental period to calculate the average daily gain (ADG). Feed intake was measured for each individual pen, and feed efficiency was determined by dividing the ADG by the average daily feed intake (ADFI) over 60 days (gain/intake). Additionally, ADFI was calculated by subtracting the feed remaining in the feeder from the feed offered.

#### 2.3.4. Blood Metabolites

At 60 days, blood samples were collected from 10 mL vacuum tubes containing K_3_EDTA (Becton Dickinson Vacutainer Systems, Franklin Lakes, NJ, USA) and then centrifuged at 3000× *g* for 15 min to separate the serum. Serum immunoglobulin G (IgG) and growth hormone levels were determined using commercial enzyme-linked immune sorbent assay (ELISA) kits. The plasma concentration of blood urea nitrogen (BUN) was determined using a UREA/BUN kit (Roche, Basel, Switzerland). An enzymatic kinetic assay was used to determine the plasma glucose concentration (GLU Kit; Roche, Mannheim, Germany).

#### 2.3.5. Fecal Microflora

To measure LAB, *Salmonella enterica*, and *E. coli* loads, fecal samples (200 g) were collected monthly from each pen at five random locations and immediately analyzed. Each fecal sample (10 g) was weighed and placed in a stomacher bag containing 90 mL of sterile saline (0.9%) at a dilution of 1:10. Fecal samples were then plated on Difco MRS agar (Difco, Detroit, MI, USA), DifcoTM SS agar (Becton, Dickinson and Company, Sparks, MD, USA), and DifcoTM Violet Red Bile agar (Becton, Dickinson and Company, Sparks, MD, USA). The MRS agar plates were incubated in a CO_2_ incubator (Thermo Scientific, Waltham, MA, USA) at 30 °C for 48 h, whereas the SS agar and Violet Red Bile agar plates were incubated for 48 h at 37 °C in an aerobic incubator (Johnsam Corp., Boocheon, Republic of Korea). Visible colonies from the plates were counted, and the number of CFU/g of fecal extract at weeks 0, 30, and 60 was calculated. Microbiological data were transformed to log10.

#### 2.3.6. Carcass Characteristics

At the end of the feeding trial, all animals were moved to the Goryeong Nonghyup Meat Processing Facility, Goryeong, Republic of Korea, and slaughtered as approved by the Ministry of Agriculture, Food, and Rural Affairs after 24 h rest. Subsequently, all cold carcasses were chilled at 2 °C for 24 h, and then, carcass characteristics (carcass weight, back fat thickness, and carcass quality grade) were measured according to the guidelines of the Animal Products Grading Service, Republic of Korea [24].

### 2.4. Statistical Analysis

Data were analyzed using analysis of variance (ANOVA) in the generalized linear model (GLM) procedure of SAS (Statistical Analysis System, version 8.2, [25]), followed by Tukey’s test to identify differences among the treatments. Significant effects were set at *p* < 0.05 and < 0.1 as tendencies. The IML procedure in SAS was used to generate linear and quadratic orthogonal polynomial coefficients for the unequally spaced data in the experiment. When a polynomial contrast (linear and quadratic effects) was significant, the effects of increasing the microbial additive supplementation levels were used.

## 3. Results

### 3.1. Growth Performance 

During the 60-day experimental period, the ADFI decreased linearly (*p* = 0.017) in the microbial additive supplementation groups, which was lower than in the control group (Table 3). In addition, ADG and feed efficiency increased linearly (*p* = 0.011 and 0.015, respectively) with increasing levels of microbial additives (*p* < 0.05). No significant differences in the initial and final weights (*p* > 0.05) were observed among the treatments.

### 3.2. Blood Metabolites

Regarding blood metabolites, the blood glucose concentration was the highest with 1.0% supplementation (*p* = 0.046, Table 4). In addition, IgG, BUN, and blood glucose levels increased linearly with increasing levels of microbial additives (*p* = 0.031, 0.049, and 0.003, respectively). No significant differences in the concentration of growth hormone were observed among the treatments (*p* = 0.212).

### 3.3. Fecal Microflora 

Fecal *Salmonella* was not detected in any of the treatments during the 60-day experimental period (Table 5). On days 30 and 60, fecal LAB counts increased linearly (*p* = 0.015 and 0.036, respectively) with increasing levels of microbial additives, whereas fecal *E. coli* counts decreased linearly (*p* = 0.048 and 0.039, respectively) with increasing levels of microbial additives.

### 3.4. Carcass Characteristics

Regarding carcass characteristics, we observed that the “1+” carcass quality grade was higher in the microbial additive supplementation groups (0.5% and 1%) than in the control group (Table 6). In addition, there were no significant differences in carcass weight or back-fat thickness among the treatments at 60 d (*p* = 0.637).

## 4. Discussion

ADG, ADFI, and F:G ratio are vital parameters for assessing performance during the pig growth phase [26]. In this study, the use of microbial additives was shown to have significant effects on pig growth performance, suggesting the beneficial effects of currently used probiotic formulations. In other words, growing–finishing pigs supplemented with probiotics demonstrated greater body weight, ADG, and feed efficiency or lower ADFI than pigs in the control group who were not supplemented with probiotics. Chen et al. [11] reported increased ADG in growing pigs fed diets supplemented with 0.2% bacillus-based probiotics. Similarly, Jeon et al. [27] reported increased ADG and feed efficiency in growing pigs fed a probiotic-supplemented diet. According to several studies, complex probiotics positively affect the growth performance of growing–finishing pigs [8,10]. Our results are consistent with those of San Andres et al. [18] and Hong et al. [28], who reported that pigs fed multi-species microbial additive diets had significantly increased ADG, and during days 28 to 35 after weaning, the use of prebiotic mixtures improved the growth performance of nursery pigs. Giang et al. [14] reported that adding a mixture of probiotics (LAB complex, *Bacillus*, and *Saccharomyces*) increased ADG and improved feed efficiency compared with the control. Notably, the above-mentioned microbial complex also has probiotic potential in growing to finishing pigs. Thus, microbial additives improve daily gain and feed efficiency owing to the digestive enzymes and growth factors derived from probiotics. For example, the addition of direct-fed microbes, commonly known as probiotics, to swine feed can improve gut health by changing the microflora environment that suppresses pathogens. Additionally, it results in increased nutrient digestibility, improved health status, and the improved growth performance of pigs [14,29,30]. The beneficial effects of prebiotics in pigs have been linked to their increased fermentability. This occurs due to apoptosis in the small intestine, which leads to increased intestinal cell proliferation, subsequently improving digestive and absorptive capacities [31,32]. The growth of weaning pigs depends on the abundance of LAB and *Bifidobacteria* [33,34]. These bacteria and their fermentation products (short-chain fatty acids and polyamines) represent the energy supply for colonic epithelial cells, aiding absorption [33,34,35]. As mentioned above (probiotics), beneficial microbes (such as LAB) in prebiotics can produce bacteriocins, lactic acid, and other compounds that improve the intestinal environment and may inhibit the growth of certain pathogens [36]. According to Liu et al. [37], using 100 or 200 mg/kg of chito-oligosaccharide (derived from chitosan) in diets improved the growth performance and digestibility of dietary nutrients in weaning pigs. However, these positive effects of probiotics may be attributed to differences in the bacterial species used in the microbial additive preparations and pig genotypes [38].

Notably, the addition of the 1% microbial extract resulted in the highest IgG concentration. This plays a major role in antibody-mediated defense mechanisms [39,40] and suggests that IgG is more important for development than the other blood metabolites in this study. Probiotics control the production of lymphocyte cytokines and exert a major effect on the immune system [41]. Cho et al. [42] observed that microbial supplementation directly added to pig diets may also cause a decrease in immune stimulation by reducing pro-inflammatory cytokines in enterocytes. Therefore, an immune change can shift the energy utilized in excessive immune stimulation toward growth and improve feed efficiency. Furthermore, our results are well supported by those of a previous study in which growing-to-finishing pigs that received supplementation with *B. subtilis* had a positive impact on the evident increase in the effect of the probiotic on IgG [43]. However, there were no differences in serum IgA and immunoglobulin M levels between the groups. Similarly, Wang et al. [44] showed that a combination of *B. subtilis* and *Enterococcus faecium* in sow diets increased serum IgG levels. This implies that an increasing IgG concentration results in a better immune response and health in growing to finishing pigs. However, the growth hormones in this study did not produce the expected results because their content in all treatments was similar, suggesting no considerable effect on the growth hormones of pigs during the growing to finishing period. Growth hormone is an important factor that primarily regulates animal growth through related receptors and downstream pathways [45]. Significant correlations between growth levels and increased weight have been reported based on animal data [46]. In this study, an increase in BUN and blood glucose values with microbial addition compared to the controls was not observed in growing pigs. Higher BUN levels represent lower nitrogen absorption efficiency, indicating an increase in lean body mass [47]. BUN levels generally decrease when the protein mass and absorption are reduced [47]. Duan et al. [48] reported that the grower phase, the control group (0%), had significantly lower BUN values than the 0.1% and 0.3% *Lactobacillus lactis* groups, whereas no difference was observed in the BUN values among the three groups for the finisher phase. For example, in the digestive tract, probiotics increase ammonia fixation and alleviate decreases in amino acid availability, which can be reduced by increasing the concentration of blood urea [49]. One observation with supplementary microbial additives at the 1% level was an increase in blood glucose concentrations. Thus, higher blood glucose levels might be explained by the activity of digestive enzymes from the microbes used or a response to increased energy absorption in the intestine [50,51]. Balasubramanian et al. [1] and Devi and Kim [52] found that microbial additive supplementation (0.1 or 0.2 g/kg multi-species probiotic, 0.2% medium-chain fatty acids, and 0.1% probiotic) had a significant effect on pig blood glucose concentrations. In contrast, Chen et al. [9] reported that feeding pigs with microbial additives (0.1 and 0.2% complex probiotics) did not affect their blood traits. However, this was not the case in the present study. At present, the mechanisms underlying these blood parameters remain unclear. In addition, blood glucose and BUN levels were within the reference ranges [53]. In terms of the effect of microbial additives on gastrointestinal health, enhancement in the ability of growing–finishing pigs to digest and ferment nutrients may correspond to an increase in the growth performance associated with immune system stimulation, including a decrease in pathogenic bacteria [44].

In this study, we determined the effects of microbial supplementation on the fecal microflora of pigs (Table 5). The increased fecal LAB or reduced fecal *E. coli* after microbial additive supplementation compared to the control is in line with the findings of Balasubramanian et al. [13], who suggested that a microbial additive containing 0.01 and 0.02% *Bacillus* spp. in basal diets affected fecal LAB counts and inhibited fecal *E. coli* counts. This may be partly explained by the presence of LAB, which are excellent antibacterial agents that suppress the growth of pathogenic microorganisms. Similar findings were reported by Lu et al. [54], who noted that supplementing the diet with a probiotic complex altered the bacterial community in the feces of weaned piglets. A study on the inclusion of multi-probiotics was reported by Giang et al. [14] in that the results of increased fecal LAB count and decreased fecal *E. coli* count in growing pigs owed to the inhibition of pathogenic microbial growth and activity by the probiotic characteristics. In addition, it has been reported that probiotics with *Bacillus* strains can not only change intestinal bacteria through colonization but can also produce specific bacteriocins by inhibiting the widest range of pathogenic bacteria [55]. In general, *Lactobacillus* spp. in probiotics can induce beneficial enzyme activities, such as sucrase, lactase, and tripeptidase in the pig small intestine and thereby promote the growth of “good” bacteria through their functions that help the absorption of nutrients and keep the balance of the intestinal or fecal microbiota [56]. Based on this information, this could be a probable reason to support our results on fecal microbes. Surprisingly, no fecal *Salmonella* in pig manure was detected in any of the treatments despite the antibacterial activity linked to the pig gut.

Furthermore, microbial supplementation resulted in no significant differences in carcass characteristics, indicating that no noticeable changes in carcass characteristics were observed during the 60-day experimental period. Exceptionally, pigs supplemented with microbial additives tended to have slight increases in carcass weight and back-fat thickness at the 1% level compared to the other groups. Junka et al. [57] and Ganeshkumar et al. [58] observed a significantly increased carcass weight in pigs that received probiotic supplementation. Chu et al. [59] reported that the carcass weight decreased in pigs fed diets supplemented with microbial additives. Because of this back-fat thickness, our observations were not in accordance with those of previous studies. Grela et al. [60] found an effect of prebiotics on back-fat thickness, which was lower in pigs fed dried Jerusalem artichokes. Other results reported by Chang et al. [61] stated that probiotic treatment groups had no significant effect on backfat thickness in pigs. Consequently, the outcomes of these studies may have been attributed to the different concentrations of microbial additives used or various important factors, such as the composition and form of the feed, interactions with probiotics, or probiotic strains [62]. Among the carcass characteristics, our data showed a higher “1+” carcass quality grade by increasing the microbial additive amount. These findings are consistent with those of a previous study, which reported that supplementing growing–finishing pig diets with *Bacillus* spp. probiotics increased meat carcass quality grade [13]. Min et al. [63] observed no beneficial effects on carcass quality grade in growing–finishing pigs fed a dietary mixture of proteases and probiotics. The discrepancies between the results of our study and those of previous studies may be due to differences in microbial abilities. However, further studies are required to evaluate the exact mechanisms of microbial action on carcass grade.

## 5. Conclusions

In conclusion, this study provides an extensive investigation of the growth performance, blood metabolites, fecal microflora, and carcass characteristics of growing to finishing pigs fed diets supplemented with microbial additives. The results show that dietary supplementation with 1.0% microbial additive effectively improved the growth performance (ADG and feed efficiency) and IgG content of the growing to finishing pigs. In addition, the 1.0% dietary microbial additives boosted the fecal microflora environment by increasing fecal LAB levels and decreasing fecal *E. coli* counts. In particular, among the carcass characteristics, these results gained a higher “1+” carcass quality grade by increasing the microbial additive, which may be due to differences in the ability of the microbials used. This study contributes to our knowledge of sustainable manure management techniques by offering valuable insights into the optimization of microbial additive levels.

## Figures and Tables

**Table 1 animals-14-01268-t001:** Microbial counts of dietary additives used in this study (log10 CFU/g).

Item	Microbial Additive
Lactic acid bacteria	7.98
*Bacillus subtilis*	7.94
Yeast	8.09

**Table 2 animals-14-01268-t002:** Ingredients and chemical compositions of basal diets (DM basis).

Item	Basal Diet
Ingredient, %	
Corn	48.5
Soybean meal	31.9
Rice bran	5.00
Tallow	4.80
Lupine	3.20
Molasses	3.00
Calcium phosphate	1.60
Lysine	0.50
Methionine	0.50
Sodium chloride	0.30
Mineral premix ^1^	0.60
Vitamin primix ^2^	0.10
Total	100.0
Chemical compositions ^3^	
ME, kcal/kg	3100
Dry matter, %	87.4
Crude protein, %	18.8
Ether extract, %	9.59
Crude ash, %	7.62

^1^ One kilogram of the diet contained the following: Fe, 70 mg; Cu, 50 mg; Zn, 25 mg; Mn, 30 mg; I, 0.7 mg; Co, 0.5 mg; Se, 0.26 mg. ^2^ One kilogram of the diet contained the following: vitamin A, 16,000 IU; vitamin D_3_, 3000 IU; vitamin E, 40 IU; vitamin B1, 2.5 mg; vitamin B2, 20 mg; vitamin B6, 4 mg; vitamin B12, 0.076 mg; vitamin K3, 2.5 mg; panthothenic acid, 40 mg; niacin, 75 mg; biotin 0.15 mg; folic acid, 0.65 mg; ethoxyquin, 12 mg. ^3^ Values represent the results of three samples, each assayed in triplicate.

**Table 3 animals-14-01268-t003:** Effects of microbial additive supplementation on the growth performance of growing–finishing pigs.

Item	Supplement, % ^1^			SEM	*p*-Value	Contrast	
	0	0.5	1.0			Linear	Quadratic
Initial weight, kg	58.5	59.0	58.0	1.322	0.640	0.595	0.425
Final weight, kg	100.6	104.0	103.5	3.589	0.440	0.286	0.558
Average daily feed intake, kg/d	1.84 ^a^	1.77 ^b^	1.73 ^b^	0.057	0.046	0.017	0.044
Average daily gain, kg/d	0.70 ^b^	0.75 ^a^	0.76 ^a^	0.041	0.011	0.011	0.343
Feed efficiency (Gain:intake)	0.38 ^b^	0.42 ^a^	0.44 ^a^	0.032	0.033	0.015	0.131

^1^ Supplemented microbial additive at 0, 0.5, and 1.0% of basal diet. ^a,b^ Means in the same row with different superscripts differ significantly (*p* < 0.05).

**Table 4 animals-14-01268-t004:** Effects of microbial additive supplementation on the blood metabolites of growing–finishing pigs.

Item	Supplement, % ^1^			SEM ^2^	*p*-Value	Contrast	
	0	0.5	1.0			Linear	Quadratic
IgG, mg/mL	21.5	22.4	23.9	2.415	0.854	0.031	0.265
Growth hormone, ng/mL	0.22	0.23	0.24	0.097	0.212	0.858	0.626
Blood urea nitrogen, mg/dL	16.0	16.4	17.5	1.683	0.331	0.049	0.144
Blood glucose, mg/dL	63.1 ^b^	63.2 ^b^	67.9 ^a^	2.356	0.046	0.003	0.101

^1^ Supplemented microbial additive at 0, 0.5, and 1.0% of basal diet. ^2^ SEM, standard error of the mean. ^a,b^ Means in the same row with different superscripts differ significantly (*p* < 0.05).

**Table 5 animals-14-01268-t005:** Effects of microbial additive supplementation on the fecal microflora of growing–finishing pigs.

Day	Microflora	Supplement, % ^1^			SEM ^2^	*p*-Values	Contrast	
		0	0.5	1.0			Linear	Quadratic
0 day	LAB ^3^	6.21	6.28	6.38	0.303	0.807	0.071	0.584
	*Salmonella*	ND ^4^	ND	ND	N/A ^5^	N/A	N/A	N/A
	*E. coli*	4.18	4.08	3.97	0.589	0.494	0.068	0.777
30 day	LAB	6.68 ^b^	7.03 ^a^	7.15 ^a^	0.124	0.026	0.015	0.584
	*Salmonella*	ND	ND	ND	N/A	N/A	N/A	N/A
	*E. coli*	4.13 ^a^	3.90 ^b^	3.74 ^c^	0.089	0.044	0.048	0.777
60 day	LAB	6.72 ^b^	6.88 ^ab^	7.09 ^a^	0.203	0.046	0.036	0.909
	*Salmonella*	ND	ND	ND	N/A	N/A	N/A	N/A
	*E. coli*	4.07 ^a^	3.92 ^b^	3.86 ^b^	0.069	0.039	0.039	0.163

^1^ Supplemented microbial additive at 0, 0.5, and 1.0% of basal diet. ^2^ SEM, standard error of the mean. ^3^ LAB, lactic acid bacteria. ^4^ ND, not detected. ^5^ N/A, not applicable. ^a–c^ Means in the same row with different superscripts differ significantly (*p* < 0.05).

**Table 6 animals-14-01268-t006:** Effects of microbial additive supplementation on the carcass characteristics of growing–finishing pigs.

Item	Supplement, % ^1^			SEM ^2^	*p*-Value	Contrast	
	0	0.5	1.0			Linear	Quadratic
Carcass weight, kg	78.2	78.3	79.7	4.801	0.756	0.160	0.474
Back-fat thickness, mm	19.5	19.1	20.5	3.478	0.637	0.082	0.157
Carcass quality grade, % (1+:1:2)	7:15:78	13:17:70	15:27:58	N/A ^3^	N/A	N/A	N/A

^1^ Supplemented microbial additive at 0, 0.5, and 1.0% of basal diet. ^2^ SEM, standard error of the mean. ^3^ N/A, not applicable.

## Data Availability

The data presented in this study are available from the corresponding author upon request.

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
