# Peer review of "The Effects of Microbial Additive Supplementation on Growth Performance, Blood Metabolites, Fecal Microflora, and Carcass Characteristics of Growing–Finishing Pigs"

_animals, 2024, doi:10.3390/ani14091268_

Round 1

Reviewer 1 Report

Comments and Suggestions for Authors

The subject of the Manuscript is interesting, but it requires rewording because it contains an incorrect approach to the experimental design and, therefore, to statistical calculations.

 One of the main remark of the Manuscript concerns the statistical model. Statistical analysis should be performed according to the two-factorial design with the Tukey test (the experiment was carried out on males and females). It can be also used one-way Anova procedure but separately for males and females.

Authors state that they performed growth performance parameters individually for each pig (l. 160-163), so re-statistical calculations, taking into account gender, should not be a problem. Please recalculate the examined indicators in a two-factorial design or separately (divided by gender) and revise the Manuscript.

Author Response

Thanks for your constructive comments. Your comments helped us to improve this manuscript. The responses to comments are mentioned below:

The subject of the Manuscript is interesting, but it requires rewording because it contains an incorrect approach to the experimental design and, therefore, to statistical calculations.

 One of the main remark of the Manuscript concerns the statistical model. Statistical analysis should be performed according to the two-factorial design with the Tukey test (the experiment was carried out on males and females). It can be also used one-way Anova procedure but separately for males and females.

Authors state that they performed growth performance parameters individually for each pig (l. 160-163), so re-statistical calculations, taking into account gender, should not be a problem. Please recalculate the examined indicators in a two-factorial design or separately (divided by gender) and revise the Manuscript.

Response: The animals in this study were not divided by gender. This study was conducted with a randomized experimental design but not with a two-factorial design. The “mixed sex” in the text means that males and females were housed in the pen together. Now, we deleted “mixed sex” from the text because it could confuse the readers.

Reviewer 2 Report

Comments and Suggestions for Authors

The objective of the study was to evaluate the effect of diet supplementation with microbial aditives on growth performance, blood metabolites, fecal microflora and carcass characteristics of pigs. That manuscript is in scope of journal, however it needs many changes and explanation before acceptation for publication. Below you can find my suggestion for consideration:

1.      Introduction: Please, focus on discription of microbial additives and adventages of its using . In my opinion detail informations on all types of additives are not necessary.

2.      All tables: Please provide p values for main effect of supplementation.

3.      Table 1: please ise corn instead of ground corn

4.      Page 3, line 122: please correct the name of bacteria strain.

5.      Table 2 does not corespond with section 2.1.

6.      Please, chech if all abbreviations are explained.

7.      Section 2.3.1: Gross energy content can be determined by bomb calorimeter, but not digestible energy content. It was calculated form gross energy. However, Table 1 contained values of ME. Please, correct.

8.      In section 2.4. it is mentioned that cubic orthogonal polynomial contract has been evaluated. Results are laking.

9.      Table 6: Please explain what is „Grade, %”.

10.   Table 5: plase uniform the names for microflora. Please, add the information on results expression.

Author Response

Thanks for your constructive comments. Your comments helped us to improve this manuscript. The responses to comments are mentioned below:

The objective of the study was to evaluate the effect of diet supplementation with microbial aditives on growth performance, blood metabolites, fecal microflora and carcass characteristics of pigs. That manuscript is in scope of journal, however it needs many changes and explanation before acceptation for publication. Below you can find my suggestion for consideration:

  1. Introduction: Please, focus on discription of microbial additives and adventages of its using . In my opinion detail informations on all types of additives are not necessary.

Response: Thanks for your constructive comments. We modified it now.

  1. All tables: Please provide p values for main effect of supplementation.

Response: We provided it now.

  1. Table 1: please ise corn instead of ground corn

Response: We changed it now.

  1. Page 3, line 122: please correct the name of bacteria strain.

Response: We corrected it now.

  1. Table 2 does not corespond with section 2.1.

Response: We modified it now.

  1. Please, check if all abbreviations are explained.

Response: We checked it now.

  1. Section 2.3.1: Gross energy content can be determined by bomb calorimeter, but not digestible energy content. It was calculated form gross energy. However, Table 1 contained values of ME. Please, correct.

Response: We corrected it now.

  1. In section 2.4. it is mentioned that cubic orthogonal polynomial contract has been evaluated. Results are laking.

Response: We corrected it now.

  1. Table 6: Please explain what is „Grade, %”.

Response: We corrected it now.

  1. Table 5: please uniform the names for microflora. Please, add the information on results expression.

Response: We corrected it now.

Reviewer 3 Report

Comments and Suggestions for Authors

Author Response

Thanks for your constructive comments. Your comments helped us to improve this manuscript. The responses to comments are mentioned below:

Effects of Microbial Additive Supplementation on Growth Performance, Blood Metabolites, Fecal Microflora, and Carcass Characteristics of Growing-finishing Pigs

The discussion part of the article is well organized. However, the introduction part needs to modify to focusing only on the microbial additives.

Line 69-81. Authors should not focus on the additives other than microbial additive based on the aim of the study.

Response: Thanks for your constructive comments. We modified it now.

Line 111-112. Rephrase the sentence.

Response: We rephrased it now.

Line 122. 9.0 log10CFU/g>9.0 log10CFU/g. Please rectify throughout the manuscript.

Response: We rectified it now.

Line 267. weanling>weaning

Response: We corrected it now.

Line 275-276. “these contradictory?” Please clarify this sentence.

Response: We modified it now.

Line 328-331. Please rephrase for clarification.

Response: We modified it now.

Reviewer 4 Report

Comments and Suggestions for Authors

In the manuscript entitled: “Effects of Microbial Additive Supplementation on Growth Performance, Blood Metabolites, Fecal Microflora, and Carcass Characteristics of Growing-finishing Pigs”, the authors assessed the effects of microbial additives on the growth performance, blood metabolites, fecal microflora, and carcass characteristics of growing-finishing pigs. The experimental results and discussion are adequate; however, several inadequate points or misunderstandings remain throughout the paper.

Major comments:

Comment 1: The research background of the microbial additives used in this study needs to be clarified; the biology of the probiotics needs to be described; are they acid-tolerant? Is it capable of intestinal colonization?

Comment 2: Is the probiotic preparation purchased from Big Biogen the microbial additive used in this study? No other commercial probiotic preparation controls were set up in this study.

Comment 3: In the results section, no significant differences in the initial and final weights were observed among the dietary microbial additive treatments, and the ADG and feed efficiency increased. However, the pig slaughter rate was reduced. Its cost and benefits should be added to the discussion.

Comment 4: Only IgG, growth hormone, BUN, and the plasma glucose concentration were detected in the blood metabolism index. Are antibodies blood metabolites? What is the basis for selecting these four indicators? Are these four indicators representative of the blood metabolome? Is the title appropriate?

Comment 5: Only LAB, Salmonella enterica, and Escherichia coli were detected in the Fecal microflora index. What is the basis for selecting these three indicators? Why not detect Bacillus subtilis, saccharomyces cerevisiae, bacillus amyloliquefaciens, or Clostridium? Are these three indicators representative of the Fecal microflora? 

Minor comments

Comment 1: Fecal samples were collected weekly, but why are only day 30 and day 60 fecal flora measurements listed in the results?

Comment 2: Carcass grade results are not presented in the abstract, and their evaluation methods are not described in the Materials Methods section.

Comment 3: Digestive enzyme activity should be tested after microbial additives are added to animals.

Comment 4: Line 21, which led to a higher average daily feed intake,is inconsistent with the results; please check and revise.

Comment 5: The level of IgA in the intestines should be detected.

Comment 6: In this study, the meat quality of pigs was not detected.

Comment 7: Line 149, placed in180 mL should be revised.

Author Response

Thanks for your constructive comments. Your comments helped us to improve this manuscript. The responses to comments are mentioned below:

In the manuscript entitled: “Effects of Microbial Additive Supplementation on Growth Performance, Blood Metabolites, Fecal Microflora, and Carcass Characteristics of Growing-finishing Pigs”, the authors assessed the effects of microbial additives on the growth performance, blood metabolites, fecal microflora, and carcass characteristics of growing-finishing pigs. The experimental results and discussion are adequate; however, several inadequate points or misunderstandings remain throughout the paper.

Major comments:

Comment 1: The research background of the microbial additives used in this study needs to be clarified; the biology of the probiotics needs to be described; are they acid-tolerant? Is it capable of intestinal colonization?

Response: Thanks for your great comments. We tried to improve the preparation of dietary additive used in this study but not each microbe. Since we did not analyze the microbial activities, it was limited to add further information of the microbes. But some of these microbes were used in our previous study and cited in the text.  

Comment 2: Is the probiotic preparation purchased from Big Biogen the microbial additive used in this study? No other commercial probiotic preparation controls were set up in this study.

Response: We modified it now

Comment 3: In the results section, no significant differences in the initial and final weights were observed among the dietary microbial additive treatments, and the ADG and feed efficiency increased. However, the pig slaughter rate was reduced. Its cost and benefits should be added to the discussion.

Response: Thanks for your comments. There was no difference in the slaughter rate. Regarding its cost and benefits, we agree with your comments. However, we did not analyze it in this study. We may have to explore it in our future study. 

Comment 4: Only IgG, growth hormone, BUN, and the plasma glucose concentration were detected in the blood metabolism index. Are antibodies blood metabolites? What is the basis for selecting these four indicators? Are these four indicators representative of the blood metabolome? Is the title appropriate?

Response: Thanks for your comments. We tried to analyze the blood metabolites related to nutrient digestibility (Blood glucose and BUN), growth performance (growth hormone), and immunes (IgG).

Comment 5: Only LAB, Salmonella enterica, and Escherichia coli were detected in the Fecal microflora index. What is the basis for selecting these three indicators? Why not detect Bacillus subtilis, saccharomyces cerevisiae, bacillus amyloliquefaciens, or Clostridium? Are these three indicators representative of the Fecal microflora? 

Response: Thanks for your comments. An analysis of the other microbes could give us a further understanding of the effects of microbial additives used in this study. The three indicators are not representative of the fecal microflora. However, lactic acid bacteria, which represent beneficial bacteria, and Escherichia coli and Salmonella, which represent pathogenic bacteria, were designated as indicators in this study.

Minor comments:

Comment 1: Fecal samples were collected weekly, but why are only day 30 and day 60 fecal flora measurements listed in the results?

Response: It was a typo. We corrected it now.

Comment 2: Carcass grade results are not presented in the abstract, and their evaluation methods are not described in the Materials Methods section.

Response: We modified it now

Comment 3: Digestive enzyme activity should be tested after microbial additives are added to animals.

Response: Yes. We agree with your comments. However, we purchased the microbial additives from the company. We did not have the analytical protocol for the digestive enzyme activity. We may have to explore this analysis in our future study  

Comment 4: Line 21, “which led to a higher average daily feed intake,” is inconsistent with the results; please check and revise.

Response: We delate it now.

Comment 5: The level of IgA in the intestines should be detected.

Response: Thanks for your constructive comment. We agree with your comment and will analyze it in our future study. 

Comment 6: In this study, the meat quality of pigs was not detected.

Response: Thanks for your comments. We just analyzed carcass characteristics in this study.

Comment 7: Line 149, “placed in180 mL” should be revised.

Response: We changed it now.

Reviewer 5 Report

Comments and Suggestions for Authors

Dear editors and authors, 

Through review, the manuscript is well presented. On the basis of experimental research, the author improved the background information and discussion, making the research more supportive and in-depth analysis. But there is still a small problem.

1. Lines 177-178, the dilution ratio of 10g stool sample dissolved in 100ml sterile saline seems to be 1:11. Please check this data.

After these revisions, I think the manuscript is ready for the journal.

best wishes

Author Response

Thanks for your constructive comments. Your comments helped us to improve this manuscript. The responses to comments are mentioned below:

Through review, the manuscript is well presented. On the basis of experimental research, the author improved the background information and discussion, making the research more supportive and in-depth analysis. But there is still a small problem.

  1. Lines 177-178, the dilution ratio of 10g stool sample dissolved in 100ml sterile saline seems to be 1:11. Please check this data.

Response: We have corrected it now.

Round 2

Reviewer 2 Report

Comments and Suggestions for Authors

.

Reviewer 4 Report

Comments and Suggestions for Authors

The authors have revised the article.